# ElasticMVS: Learning elastic part representation for self-supervised multi-view stereopsis

**Jinzhi Zhang**[1,2*]**, Ruofan Tang**[1,3]**, Zheng Cao**[4]**, Jing Xiao**[5]**, Ruqi Huang**[2§] **and Lu Fang**[1§]
[1]Department of Electronic Engineering, Tsinghua University
[2]Tsinghua Shenzhen International Graduate School
[3]Dept. of Automation, Tsinghua University, [4]BirenTech Research, [5]Pingan Group

*zjz22@mails.tsinghua.edu.cn

## Abstract

Self-supervised multi-view stereopsis (MVS) attracts increasing attention for learning dense surface predictions from only a set of images without onerous ground-truth 3D training data for supervision. However, existing methods highly rely on the local photometric consistency, which fail to identify accurately dense correspondence in broad textureless or reflectant areas. In this paper, we show that geometric proximity such as surface connectedness and occlusion boundaries implicitly inferred from images could serve as reliable guidance for pixel-wise multi-view correspondences. With this insight, we present a novel elastic part representation, which encodes physically-connected part segmentations with elastically-varying scales, shapes and boundaries. Meanwhile, a self-supervised MVS framework namely ElasticMVS is proposed to learn the representation and estimate per-view depth following a part-aware propagation and evaluation scheme. Specifically, the pixel-wise part representation is trained by a contrastive learning-based strategy, which increases the representation compactness in geometrically concentrated areas and contrasts otherwise. ElasticMVS iteratively optimizes a part-level consistency loss and a surface smoothness loss, based on a set of depth hypotheses propagated from the geometrically concentrated parts. Extensive evaluations convey the superiority of ElasticMVS in the reconstruction completeness and accuracy, as well as the efficiency and scalability. Particularly, for the challenging large-scale reconstruction benchmark, ElasticMVS demonstrates significant performance gain over both the supervised and self-supervised approaches. Code is avaliable at https://thu-luvision.github.io.

## 1   Introduction

Given a set of images with known camera parameters, multi-view stereopsis (MVS) aims to reconstruct the dense and accurate geometry of the scene. Existing learning-based approaches [47], typically being cast as an end-to-end depth regression task, outperform the traditional geometry-based ones [34]. However, their generalization ability in open environments is severely hindered by the availability of the onerous laser-scanned 3D training data. Self-supervised MVS [24, 11] lifts such limitation by leveraging the multi-view photometric consistency in place of the supervisory signals. Unfortunately, the pixel-level photometric regularization is susceptible to textureless patterns and illumination variations, leading to incomplete and inaccurate reconstructions, especially in outdoor environments.

---

§Corresponding Author: ruqihuang@sz.tsinghua.edu.cn, fanglu@tsinghua.edu.cn, luvision.net

36th Conference on Neural Information Processing Systems (NeurIPS 2022).

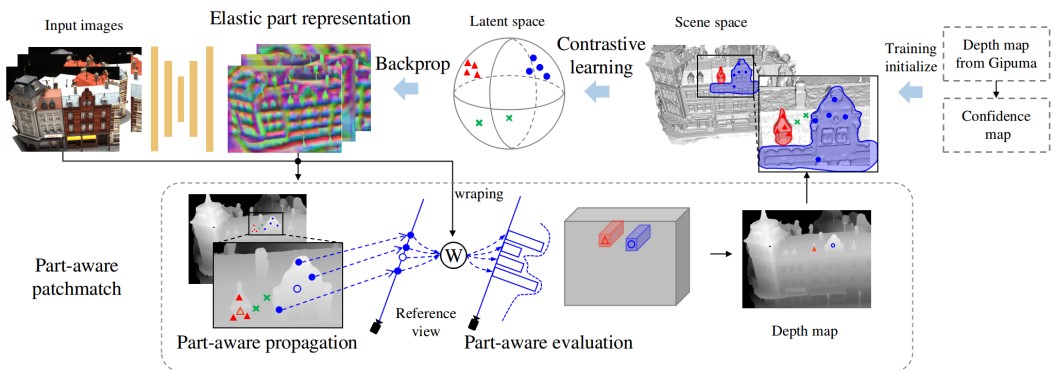

Figure 1: An overview of ElasticMVS, a self-supervised MVS framework to learn the representation and estimate per-view depth. The elastic part representation of each image is trained by a contrastive learning-based strategy, which increases the representation compactness in geometrically concentrated areas and contrasts otherwise. Benefiting from the elastic part representation, the inherent geometric correlations are effectively exploited by a patchmatch-based strategy. Without the ground-truth 3D information, positive and negative samples for training are based on the depth map from the traditional geometry-based stereopsis [34].

In this paper, we tackle the above challenges with a novel image parsing paradigm. Instead of treating images as projections of 3D scenes, we show that the inherent geometric correlations, such as surface connectedness, smoothness and boundaries, could be inferred implicitly from images and serve as a reliable guidance for pixel-wise multi-view correspondence estimation. The mainstream solutions of feature-based regularizations introduce either hand-crafted structures such as lines and planes from superpixels [32, 4, 30], or semantic features pre-trained from limited geometric-unrelated labels [43, 17]. Different from those, we aim to build the correlation between the latent space of the representation and the scene space of the surface in a self-supervised way. With this insight, we present a novel elastic part representation which encodes physically-connected part segmentations with elastically-varying scales, shapes and boundaries. Unlike the traditional segmentation networks with a fixed number of labels, our pixel-wise part representation directly reflects the surface correlation, i.e., the more similar the two representations are, the higher probability they belong to the same physical surface part. Meanwhile, a self-supervised MVS framework namely ElasticMVS is proposed to learn the representation and estimate per-view depth iteratively following a part-aware propagation and evaluation scheme. We believe that this framework provides a neat solution to bridge the gap between the multi-view stereopsis and the point cloud reconstruction [3], which has the potential to be effectively exploited by many other MVS methods.

Benefiting from the elastic part representation, the inherent geometric correlations are effectively exploited by a patchmatch-based propagation strategy [13]. Specifically, in each iteration, depth is estimated by selecting the best depth hypotheses adaptively propagated only from the geometrically concentrated part. Different from other adaptive propagation strategy [39, 44] based only on pure image features without explicit geometric constraints, our part-aware propagation scheme directly relies on physically plausible indicators. Besides, the pixel-level photometric consistency loss is enhanced by the constraints of feature-level correspondence and non-local piecewise surface smoothness, which provide part-aware matching cues to guide a more robust depth prediction in textureless and reflectant areas. To train the elastic part representation, a contrastive learning-based strategy is implemented by increasing the representation compactness in geometrically concentrated areas and contrasts otherwise. Without the ground-truth 3D information, these geometrically concentrated areas are identified by the depth map generated from the initialized ElasticMVS without the part representation, which owns the same workflow as the traditional geometry-based stereopsis [34]. A noise-robust training loss is implemented to get rid of undesirable identifications.

Experiments on 'DTU' and 'Tanks and Temples (T&T)' datasets demonstrate that the proposed ElasticMVS significantly outperforms all state-of-the-art self-supervised learning-based methods. Remarkably, in the challenging large-scale reconstruction benchmark (T&T advanced), ElasticMVS

shows dramatic performance improvements w.r.t. both the supervised and self-supervised approaches. Besides, the systematic evaluations indicate that our elastic part representation is superior in modeling elastically-varied parts, while being robust and lightweight in training and inference.

## 2 Related work

### 2.1 Multi-view Stereopsis (MVS)

The major works in multi-view stereopsis focus on the view-wise depth map estimation [36, 12] and then fuse together to form the point cloud[50, 14]. Among these methods, patchmatch-based depth optimization methods[13, 34, 44], initialized from the randomly sampled depth hypotheses, show great run-time and memory efficiency in depth map estimation. Some learning-based methods [39, 26] introduce an iterative coarse-to-fine optimization in an end-to-end trainable architecture to improve each of the core steps in patchmatch. However, these algorithms suffer from incomplete and noisy depth predictions in textureless and reflectant areas. Several previous works in MVS solved the problem by introducing pre-defined global structural priors such as planes to resolve the ambiguity. For instance, TAPAMVS [32] segments the whole image into a bunch of superpixels and assumes textureless areas piecewise planar. RANSAC is applied to estimate its corresponding plane parameters. Beyond that, a bunch of works in point cloud reconstruction deal with reconstruction artifacts with outlier, noisy or missing data. For instance, implicit representation [22] is used to provide global surface smoothness prior given noisy data, and exterior visibility [21] is used to detect and eliminate redundant data. However, neither the handcrafted plane structure nor the complex smoothness prior can provide sufficiently diverse and accurate surface part modeling for real-world scenarios.

Currently, learning methods based on 3D cost volume regularization demonstrate state-of-the-art results in several benchmarks. These methods optimizes the 3D geometry by 3D CNN, unprojecting from 2D images or features into the 3D volume [18, 19, 23, 47, 20, 48, 49, 7, 15, 10, 28, 52]. However, these supervised methods crucially rely on ground-truth 3D data, which are hard to generalize in wild open environments. To realize self-supervised MVS without the onerous ground-truth 3D training data, [11, 16, 24, 51, 43, 29, 31] replace the supervision signal with the unsupervised rendering loss or cross-view feature consistency loss. However, reprojection under multiple views is highly sensitive to environmental illumination, which is hard to generalize in highly variable scenarios.

### 2.2 Part Representation

Recently, learning representations from the image in a self-supervised training strategy has gained growing popularity in 2D vision tasks, which use contrastive learning [42, 8, 9, 6] to deal with the lack of ground-truth labels. They use InfoNCE [38] to generate a unique embedding for each picture in a batch. [40] further leverages a dense projection head to generalize such embedding from image-level to patch-level, which is more suitable for performing dense prediction tasks such as semantic segmentations. [17, 41] assigns the equivariance constraint and the semantic consistency regularization on the learned part response representation [53] to predict part segmentations that are consistent across different instances or camera poses. [43] incorporates learned semantic features into multi-view stereopsis pipeline to boost the reconstruction ability. However, previous works only concentrate on generating semantic features and using it forward to perform down-stream tasks. Instead, we utilize the rough depth map as supervision to the semantic learning process, which forms a natural integration of photometric and geometric information to provide elastic part representations.

## 3 Method

We estimate the depth maps of a given reference image $x$ and the corresponding $M$ unstructured source images with known camera calibrations. The final point cloud reconstruction is obtained by fusing all of them. Since the geometry information gained from any photometric loss contains missing data and artifacts in some areas of the image (especially in textureless regions), a heuristic geometry regularization for depth prediction is unavailable in these regions. Therefore, the key innovation in our framework is a novel *elastic part representation*, which encodes sufficient geometric details to guide a piecewise-smooth depth map prediction. We first formally define our part-aware representation in Sec. 3.1. Based upon the learned representation, we introduce two improvements on depth hypotheses

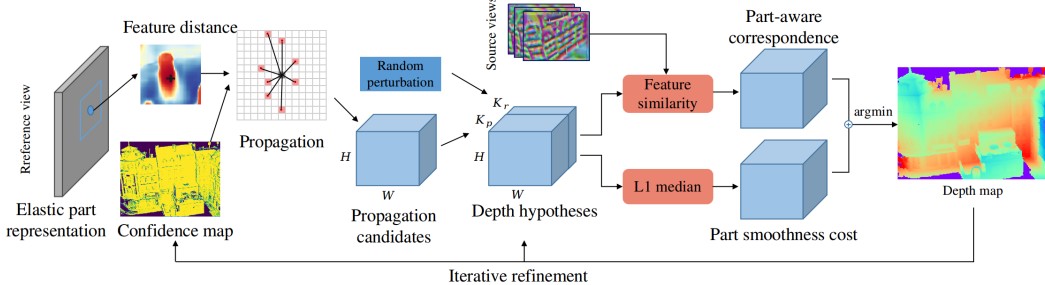

Figure 2: Detailed architecture of part-aware patchmatch. Given the pixel-wise elastic part representation from the ConvNet, depth hypotheses are sampled from high confidence regions which own similar features. The optimal depth values in each iteration are chosen based on the combination of a part-aware correspondence and a part smoothness cost.

propagation and evaluation for part-aware patchmatch in Sec. 3.2. We present the self-supervised training scheme in Sec. 3.3 to robustly train the representation with noisy initialization.

## 3.1 Elastic Part Representation

Suppose that $(\tilde{d}, \tilde{n}) \in \mathbb{R}^{H \times W \times 4}$ are the underlying depth and normal map of the image $x$, we first define the geometry-aware part segmentation $S_p$ related to each pixel location $p \in \mathbb{R}^2$, which is geometrically concentrated inside: $S_p \triangleq \{q \in \mathbb{R}^2 | \Gamma\{(\tilde{d}_p, \tilde{n}_p); (\tilde{d}_q, \tilde{n}_q)\} \leq \epsilon\}$, where $\Gamma\{(\tilde{d}_p, \tilde{n}_p); (\tilde{d}_q, \tilde{n}_q)\} \triangleq |([q, \tilde{d}_q] - [p, \tilde{d}_p]) \cdot \tilde{n}_p|$ is the 3D point-to-plane distance [35] between two pixels.

Equipped with the part segmentation $S_p$ for each pixel $p$, we propose to learn a per-pixel elastic part representation $z_p \in \mathbb{R}^D$ for each $p \in \mathbb{R}^2$, such that $z_p$ is compact in the geometric concentrated part $S_p$ and contrast otherwise. Specifically, we use the Soft-Nearest Neighbor Loss [33] to formulate such:

$$L_p = -\sum_p \log \frac{\sum_{p^+ \in S_p} \exp\left(\langle z_p, z_{p^+} \rangle / \tau\right)}{\sum_{q \neq p} \exp\left(\langle z_p, z_q \rangle / \tau\right)}, \tag{1}$$

where $z_{p^*} \in z$ is the per-pixel elastic part representation on pixel location $p^*$, $\langle \cdot, \cdot \rangle$ is the dot product, and $\tau$ is the temperature.

Finally, we use a ConvNet, $F_\Theta(\cdot)$, to model the generation of the elastic part representation, i.e., for the reference image $x \in \mathbb{R}^{H \times W \times 3}$, $F_\Theta(x) \in \mathbb{R}^{H \times W \times D}$ denotes the collection of per-pixel elastic part representations.

## 3.2 Part-aware Matching

After training the part representation (described in Sec. 3.3), we build our part-aware matching module on top of the seminal work of patchmatch [13]. That is, the final depth prediction is iteratively refined from the beginning of the random initialization. In particular, we incorporate our elastic part representation and propose the following two novel improvements: 1. Propagation: we propagate hypotheses to neighbors with our learned elastic part representation; 2. Evaluation: we evaluate and choose the best hypotheses based on photometric consistency, feature-level correspondence, and geometrically smoothness. The resulting module is differentiable, and the architecture of part-aware patchmatch is illustrated in Fig. 2.

**Propagation.** The key idea of the part-aware propagation is to gather hypotheses from the nearby pixels of the same physical surface part. As illustrated in Fig. 3, instead of propagating hypotheses naively from a static set of neighbors [13], our depth hypotheses are sampled from the nearby pixels which own similar features of the learned elastic part representation.

Since the depth value is unreliable in textureless and highly reflectant areas, a confidence map $c \in [0, 1]^{H \times W}$ is generated to pick the reliable candidates from noise, which is a combination of

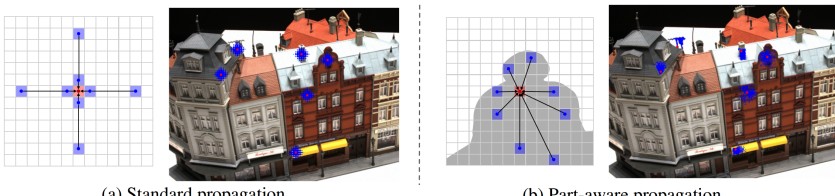

(a) Standard propagation            (b) Part-aware propagation

Figure 3: Visualization of all sampled locations for propagation (blue points) during 3 iterations of patchmatch given six pixels (red points).

the photometric consistency and the geometric consistency [34]. Please refer to the supplementary material for a more detailed explanation. For each pixel $p$ in the depth map to be updated, $K_p$ patch hypotheses are propagated from the propagation candidates $\mathcal{T}_p$, which is a set of nearby pixels $q$ whose feature $z_q$ is close to the feature $z_p$ with high confidence $c_q$:

$$\mathcal{T}_p = \left\{ q \in R^2 \middle| \|z_p - z_q\| \leq \eta, c_q \geq \xi \right\}. \tag{2}$$

According to the property of elastic part representation in Eq. 1, the generated depth hypotheses from $\mathcal{T}_p$ would stay close to the underlying surface.

**Evaluation.** Beyond the photometric consistency loss, we propose two novel losses for hypothesis evaluation – Part-aware correspondence loss and Part smoothness loss.

*Part-aware correspondence loss* evaluates the feature similarity of the elastic part representation between the reference view and the warped source views. Given in homogeneous coordinates and depth hypothesis $d$, we obtain the warped source feature $z^{[i]}_{p_i(d)}$ of the i-th source view via differentiable bilinear interpolation. Therefore, given the features $z^{[0]}_p, z^{[i]}_{p_i(d)}$ of the elastic part representation in the reference and source view respectively, the part-aware correspondence at depth $d_p$ and normal direction $n_p$ is given by:

$$M_s(d_p, n_p | x, z) = \left[ \sum_i \langle z^{[0]}_p, z^{[i]}_{p_i(d)} \rangle + \alpha_s \cdot \rho(d_p, n_p | x, x^{[i]}) \right], \tag{3}$$

where $\rho(\cdot)$ is the bilateral weighted normalized cross correlation [34] implemented in the traditional patchmatch algorithm.

*Part smoothness loss* aims to augment the local patchmatch evaluation with a non-local piecewise smoothness term corresponding to the same physical surface part. Inspired by the work of Locally Optimal Projection (LOP) [27] for surface reconstruction from point clouds, the part smoothness loss is formulated using an L1 median loss, which is robust to the outliers and gives a piecewise second-order surface approximation in each surface part [27].

More specifically, the depth map $d \in \mathbb{R}^{H \times W \times 1}$ is firstly transferred to the point set $e \in \mathbb{R}^{H \times W \times 3}$ in the scene space given the camera matrix $\mathbf{M}$: $e = \{ e_{p^*} \in \mathbb{R}^3 | [e_{p^*}^\top, 1]^\top = \mathbf{M}^{-1}[p^{*\top}, d_{p^*}, 1]^\top \}$. Then we define our part smoothness loss as the sum of the Euclidean distance to the points from propagation candidates in $\mathcal{T}_p$ (Eq. 2):

$$M_g(d_p, n_p \mid z) = \sum_{q \in T_p} \omega_q \|e_p - e_q\|, \tag{4}$$

where $\omega_q = \exp\{-c_p - \alpha_n \langle n_p, n_q \rangle\}$ is a weight correlated with the confidence $c_q$ and the normal similarity.

Putting all the components together, as illustrated in Fig. 2, the depth map initialization consists of $K_p$ random values within the depth range of interest. For subsequent iterations, $K_p$ hypotheses are propagated based on the depth values of the former one. To deliver a more diverse set of hypotheses than just using propagation, another $K_r$ hypotheses are sampled by adding a small

random perturbation to the previous estimation. In each iteration, the optimal depth and normal are chosen as: $(d_p^{\mathrm{opt}}, n_p^{\mathrm{opt}}) = \underset{d_p^*, n_p^*}{\operatorname{argmin}} \left\{ M_s(d_p^*, n_p^* | x, z) + \alpha_g \cdot M_g(d_p^*, n_p^* \mid z) \right\}$. This function is not directly solved using the first-order optimization mechanisms such as gradient descent, but rather approximated using a discrete sampling strategy, which has long been proposed in the field of stereo-matching [34]. Empirically, 5 iterations are sufficient to obtain an accurate piecewise smoothness depth map.

### 3.3 Network Training

The elastic part representation $z$, as described in Sec. 3.1, is compact in the area of a smooth surface part. We now describe how to train the ConvNet $F_\Theta(x)$ using the self-supervised contrastive learning strategy without knowing the ground-truth depth and normal information.

The core of contrastive learning is to construct a set of positive and negative samples, in which the representation in the positive samples stay close to each other while the negative ones are far apart. Before training, a depth map $d$ and a normal map $n$ of each reference image $x$ are propagated and generated by the initialized ElasticMVS without the part representation, which is similar to the traditional patchmatch-based algorithm, e.g., Gipuma [13].

When selecting positive and negative samples during the training stage, we get rid of the noise and error in the initial depth map using the confidence map. For each pixel $p$ with high confidence ($c_p \geq \xi$), we construct a set of pixel candidates $\hat{\mathcal{S}}_p$ for positive samples, in which each of them is geometrically concentrated to $p$ with high confidence:

$$\hat{\mathcal{S}}_p = \{q \in \mathbb{R}^2 | \Gamma\{(d_p, n_p); (d_q, n_q)\} \leq \epsilon, c_q \geq \xi\}. \tag{5}$$

Therefore, the dense contrastive loss for self-supervised training is defined as:

$$L_c = - \sum_p \mathbf{1}_{[c_p \geq \xi]} \log \frac{\sum_{p^+ \in \hat{\mathcal{S}}_p} \exp\left(\langle z_p, z_{p^+}\rangle / \tau\right)}{\sum_{q \neq p} \exp\left(\langle z_p, z_q\rangle / \tau\right)}. \tag{6}$$

This loss function encourages representation compactness in parts with close surface distance and separation otherwise, i.e., the distance in the representation space naturally reflects the distance in the 3D scene space. Since this loss function only acts on pixels with above-the-threshold confidence, it prevents the initial reconstruction artifacts from ruining the contrastive learning process. Due to the memory limitation, it is impossible to densely sample all positive and negative samples from the whole image in Eq. 6. Therefore, we randomly select $N_c$ points from all samples in $\hat{\mathcal{S}}_p$, using a probability distribution inversely proportional to the distance away from $p$.

To obtain spatially concentrated representation, we utilize a spatial concentration loss [46] that encourages all the pixel embeddings to become isotropically isolated:

$$L_s = \sum_p \left\| \frac{\sum_{q \in \hat{\mathcal{S}}_p} \exp\left(\beta \cdot \langle z_p, z_q\rangle\right) \cdot q}{\sum_{q \in \hat{\mathcal{S}}_p} \exp\left(\beta \cdot \langle z_p, z_q\rangle\right)} - p \right\|, \tag{7}$$

where $\beta$ is a constant parameter that controls the weight of the feature similarity between $p$ and $q$, which prompts the weighted average of the sampled neighboring points $q$ to be close to the point $p$. Therefore, the pixel locations far away from the sampled point will be given low weights, leading to a spatially concentrated representation. The concentration loss enables the spread of reasonable distance between each representation from high-confidence regions to low ones, which enables a robust training of the representation towards noisy and incomplete depth predictions. A more detailed analysis is demonstrated in Table 4 and Fig. 6.

Our final loss function is a weighted sum of contrastive loss and concentration loss, defined as :

$$L_{total} = L_c + \gamma_s \cdot L_s, \tag{8}$$

where $\gamma_s$ denotes the weight of concentration loss in the total loss function.

### 3.4 Implementation Details

The elastic part representation is trained with a Feature Pyramid Network , which shares the same structure as in PatchmatchNet [39]. To train the network, we use the SGD [5] with an initial learning

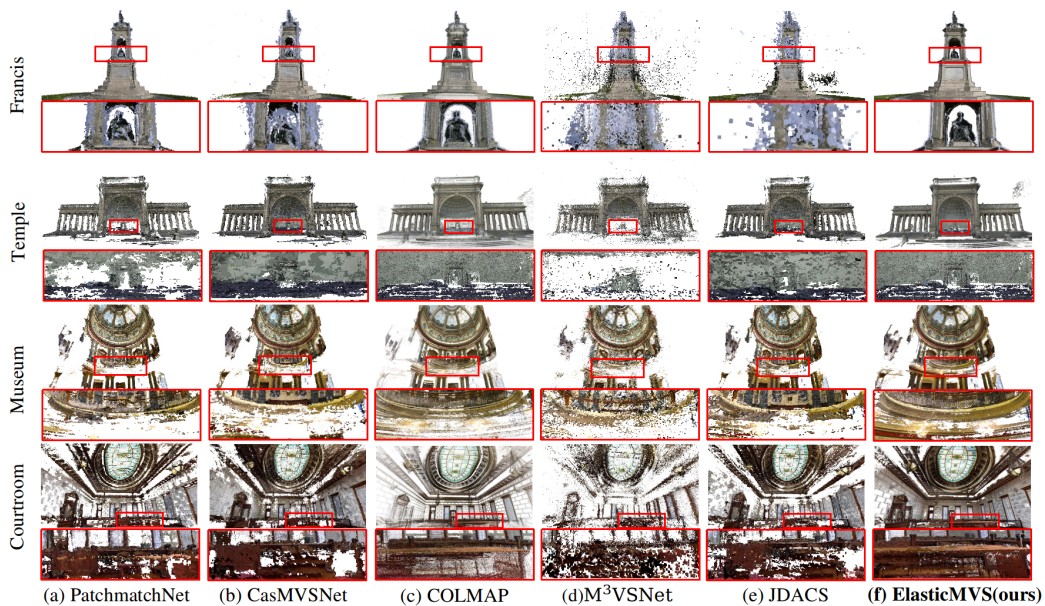

Francis

Temple

Museum

Courtroom

| (a) PatchmatchNet | (b) CasMVSNet | (c) COLMAP | (d)M³VSNet | (e) JDACS | **(f) ElasticMVS(ours)** |

Figure 4: Point cloud visualization in the T&T intermediate and advanced benchmarks, compared with three types of reconstruction methods: supervised methods (a)(b) [39, 15], self-supervised methods (d)(e) [16, 43], and one traditional geometry-based method (c) [34].

rate of $1.0 \times 10^{-4}$. The threshold $\epsilon$ for defining $S_p$ is set to $0.5$ in the DTU dataset and $0.01$ in the T&T dataset. We fix the temperature number $\tau = 0.5$ and the confidence threshold $\xi = 0.7$ during training and propagation. We sample $N_c = 8$ points for each pixel. During the propagation, we select $K_p = 8$ hypotheses for propagation and $K_r = 4$ for perturbation. We choose $\eta = 0.1$ in Eq. 2 to find nearby pixels. During the reconstruction stage, we set respectively $\alpha_s = 0.1$ and $\alpha_g = 0.5$ in Eq. 3 and 4. For the loss functions, $\beta = 0.5$ in Eq. 7 and $\gamma_s = 0.005$ in Eq. 8 All the training and reconstruction are conducted on a single NVIDIA GTX 3090 graphics card.

## 4   Experiment

### 4.1   Benchmarks

**DTU dataset [1]** is an MVS dataset consisting of different indoor objects scanned at 49 camera positions under 7 different lighting conditions. We use the original image size ($1600 \times 1200$) as input and follow the same evaluation protocol as [1] to compute the accuracy and the completeness of the final reconstructions. The overall distance metric is defined as the mean of the two metrics.

Quantitative results with three types of reconstruction methods are shown in Table 1, including geometry-based, supervised and self-supervised MVS methods. Our method outperforms all the self-supervised methods including SurRF [51] and JDACS [43] in *accuracy* and *overall* distance metrics, and achieves the second-best performance in *completeness*. It is remarkable that ElasticMVS is competitive compared to state-of-the-art supervised methods,

Table 1: Evaluation of reconstruction quality by measuring the distance metric (lower is better) on the DTU dataset [1].

| Method | Acc.(mm) | Comp.(mm) | Overall(mm) |
|---|---|---|---|
| Furu [12] | 0.613 | 0.941 | 0.777 |
| Tola [36] | 0.342 | 1.190 | 0.766 |
| Gipuma [13] | **0.283** | 0.873 | 0.578 |
| COLMAP [34] | 0.400 | **0.664** | 0.532 |
| MVSNet [47] | 0.396 | 0.527 | 0.462 |
| SurfaceNet+ [19] | 0.385 | 0.448 | 0.416 |
| Point-MVSNet [7] | 0.342 | 0.441 | 0.376 |
| CasMVSNet [15] | **0.325** | 0.385 | 0.355 |
| UCS-Net [10] | 0.338 | 0.349 | **0.344** |
| PVA-MVSNet [49] | 0.379 | 0.336 | 0.357 |
| D²HC-RMVSNet [45] | 0.395 | 0.378 | 0.386 |
| PatchmatchNet [39] | 0.427 | **0.277** | 0.352 |
| Consistency [24] | 0.881 | 1.073 | 0.977 |
| SurRF [51] | 0.388 | 0.390 | 0.389 |
| MVS² [11] | 0.760 | 0.515 | 0.637 |
| M³VSNet [16] | 0.636 | 0.531 | 0.583 |
| JDACS [43] | 0.398 | **0.318** | 0.358 |
| **ElasticMVS (ours)** | **0.374** | 0.325 | **0.349** |

e.g., our overall distance metric is the second best among *all* methods.

**Tanks and Temples [25]** dataset contains two benchmarks (*intermediate* and *advanced*) on outdoor real-world scenes with complex geometry. We use the original image size (1920 × 1080) as the input. We evaluate the results with F-score [25], the higher value of which indicates the better reconstruction. We summarize the quantitative results in Table 2. On the *intermediate* benchmark, ElasticMVS achieves state-of-the-art performance among all supervised and self-supervised MVS methods. Especially, our method outperforms all the self-supervised MVS methods by a significant margin. On the more challenging *T&T advanced* benchmark, which contains large-scale real-world scenes with more than 300 camera views for each scene, ElasticMVS even compares favourably with the supervised methods. The tremendous performance improvement in this benchmark demon-

Table 2: Quantitative evaluation (F-score, higher is better) of state-of-the-art learning-based methods on two benchmarks of the Tanks and Temples (T&T) dataset [25].

| Method | Intermediate | Advanced |
|---|---|---|
| MVSNet [47] | 43.48 | - |
| CasMVSNet [15] | **56.84** | 31.12 |
| UCSNet [10] | 54.83 | - |
| PVAMVSNet [49] | 54.46 | - |
| SurfaceNet+ [19] | 49.38 | - |
| R-MVSNet [48] | 50.55 | 29.55 |
| Point-MVSNet [7] | 48.27 | - |
| PatchmatchNet [39] | 53.15 | **32.31** |
| Patchmatch-RL [26] | 51.81 | 31.78 |
| MVS$^2$ [11] | 37.21 | - |
| M$^3$VSNet [16] | 37.67 | - |
| SurRF [51] | 54.36 | - |
| JDACS [43] | 45.48 | - |
| COLMAP[34] | 42.14 | 27.24 |
| **ElasticMVS (ours)** | **57.88** | **37.81** |

strates the effectiveness and robustness of the proposed ElasticMVS w.r.t complex categories and geometries.

We also compare our reconstructions with various baselines qualitatively in Fig. 4. ElasticMVS produces much more complete predictions compared with COLMAP [34], while generating much less noise than existing self-supervised methods (M3VSNet [16] and JDACS [43]). Besides, we achieve even surpassing performances upon state-of-the-art supervised methods (CasMVSNet [15] and PatchmatchNet [39]).

## 4.2 Ablation Study

In this section, we provide detailed ablation study and analysis of the key components of ElasticMVS, including the part-aware patchmatch and the elastic part representation.

**Part-aware patchmatch.** To investigate the effects of the key components in the part-aware patchmatch during inference, we conduct a comparison with respect to the part-aware propagation and evaluation described in Sec. 3.2. Our part-aware patchmatch is compared with three variants: 1. Propagation without elastic Part Representation (w/o PR), which employs fixed sampled locations for propagation similar to Gipuma [13]; 2. Evaluation without Part-aware Correspondence (w/o PC), which only employs the photometric consistency in Eq. 3; 3. Evaluation without Part Smoothness loss (w/o PS) in Eq. 4, which simply employs an L1-median loss from all neighborhoods, similar to the LOP [27] for point cloud de-noising.

As shown in Table 3, our part-aware propagation and evaluation modules both play important roles in boosting the overall performance across different datasets. Qualitative results are shown in Fig. 5. Implementations without the elastic part representation (a) and the part-aware correspondence (b) fail to recover the dense surface in textureless regions. Although applying a naive surface de-noising and completion method (c) helps to fill in some of the holes in textureless areas, it still suffers from low accuracy and noisy predictions due to the lack of part aware-

Table 3: Ablation study of part-aware patchmatch, without the help of part representation (w/o PR), part-aware Correspondenc (w/o PC) and part smoothness (w/o PS)

| Methods | DTU (mm) | T&T (%) |
|---|---|---|
| w/o PR | 0.455 | 40.05 |
| w/o PC | 0.430 | 43.82 |
| w/o PS | 0.411 | 39.58 |
| Ours | **0.349** | **57.88** |

ness. Overall, the strong geometric coherence revealed by the learned elastic sampling strategy helps ElasticMVS gather more accurate and complete depth maps in both small complex surfaces and large flattened areas.

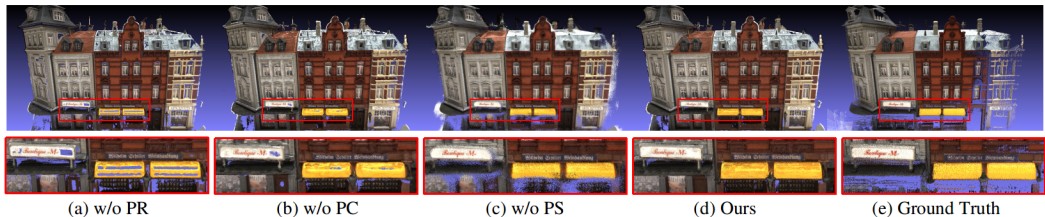

| (a) w/o PR | (b) w/o PC | (c) w/o PS | (d) Ours | (e) Ground Truth |

Figure 5: Reconstruction results of ablation study on part-aware patchmatch, without part representation (w/o PR), part-aware correspondence (w/o PC) and part smoothness (w/o PS) relatively.

**Training loss of elastic part representation.** Now we study how different training losses described in Sec. 3.3 affect the performance of the representation and further the depth map predictions. Our elastic part representation is compared with two variants, which are trained without confidence map (w/o Con.) in Eq. 6 and without spatial concentration loss (w/o Spa.) in Eq. 7.

We test on the DTU dataset. We first measure the compactness of the learned part representation as described in Eq. 1 using the ground truth depth map. We also compare the predicted depth map error using these representations, which is evaluated by the L1 distance from the predicted depth map to the ground-truth. As shown in Table 4, our elastic part representation is the most compact one (with lowest contrastive loss).

Table 4: Ablation study of the representation concerning the training loss without the help of confidence map (w/o Con.) and spatial concentration (w/o Spa.).

| Methods | Contrastive loss | Depth error(mm) |
| --- | --- | --- |
| w/o Con. | 2.52 | 3.55 |
| w/o Spa. | 2.34 | 3.01 |
| Ours | **2.28** | **2.52** |

Visualizations of the representation are demonstrated in Fig. 6, shown by a heat map of the representation distance $\|z_q - z_p\|$ from a pixel location $p$ to all other pixels $q$ in the whole image. As expected, our elastic part representation successfully recognizes the part on the same physical surface, resulting in proper surface boundary identifications with elastically-varying scales and shapes. Contrastingly, the representation trained without the confidence map (w/o Con.) fails to identify a complete surface part due to the noisy training samples. Meanwhile, the representations trained without spatial concentration loss (w/o Spa.) suffer from outliers far from the surface part, demonstrating the effectiveness of spatial regularization.

## 4.3 Discussion

**Memory.** In Table 5, we compare the GPU memory consumption and the run-time during inference with two memory-efficient learning-based methods. We choose scan 9 and a fixed reference view downsampled to the scale of $640 \times 800$ in the DTU dataset in this study. As shown in Table 5, ElasticMVS is more memory-efficient compared with two learning-based MVS backbones. Notice that the memory and running time are adaptively changed with the hypothesis number and the iteration number, since propagation with more hypotheses will converge with fewer iterations, but take a larger memory consumption. Overall, the experiment demonstrates the superiority memory efficiency of ElasticMVS.

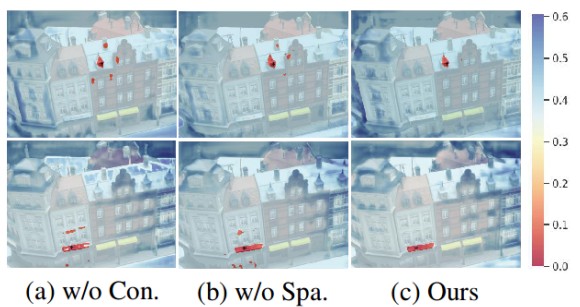

| (a) w/o Con. | (b) w/o Spa. | (c) Ours |

Figure 6: Visualization of the elastic part representation for two pixels (black). Con. denotes confidence map, and Spa. denotes spatial concentration loss.

**Impact of parameters.** To better understand the effect of different parameters, we first evaluate the reconstruction quality w.r.t different thresholds. As shown in Table 6, with lower $\epsilon$, the reconstruction accuracy grows higher while the completeness becomes lower. The reason is that $\epsilon$ (point-to-plane

Table 5: Memory and running time comparison with two learning-based MVS backbones.

| Methods | No. of stages | GPU Mem. (MB) | Run-time (s) |
|---|---|---|---|
| CasMVSNet [15] | 3 | 3489 | 0.12 |
| PatchmatchNet [39] | 5 | 4439 | **0.08** |
| ElasticMVS (8 hypotheses) | 5 | **2679** | 0.48 |
| ElasticMVS (16 hypotheses) | 3 | 5785 | 0.37 |

Table 6: Impact of parameters during inference.

| method | Acc. (mm) | Comp. (mm) | Overall (mm) |
|---|---|---|---|
| $\epsilon = 0.005, \xi = 0.7$ | **0.368** | 0.354 | 0.362 |
| $\epsilon = 0.02, \xi = 0.7$ | 0.380 | **0.301** | **0.341** |
| $\epsilon = 0.04, \xi = 0.7$ | 0.396 | 0.331 | 0.364 |
| $\epsilon = 0.01, \xi = 0.5$ | 0.540 | 0.342 | 0.441 |
| $\epsilon = 0.01, \xi = 0.6$ | 0.405 | **0.317** | 0.361 |
| $\epsilon = 0.01, \xi = 0.8$ | **0.358** | 0.330 | **0.344** |
| $\boldsymbol{\epsilon = 0.01, \xi = 0.7}$ | 0.374 | 0.325 | 0.349 |

distance) controls the compactness of the surface. Smaller $\epsilon$ during inference leads to a more strict surface boundary with smaller regions.

On the other hand, with a higher confidence threshold $\xi$, the accuracy and the overall performance of the prediction become higher. With a loose threshold ($\xi = 0.5$), the performance drops dramatically. Therefore, it is better to set the confidence threshold at a high level.

**Limitation and Future work.** It takes a long time in ElasticMVS to calculate the patch similarity for each pixel, since the total computational complexity to evaluate the photo consistency from $M$ source images of size $H \times W$ with $K$ hypotheses is $O(M \times H \times W \times K \times l^2)$, where $l$ is the patch size. Empirically, $l = 11$, which is impossible to give a forward and backward pass from a single graphics card at once. As shown in Table 5, the running time of ElasticMVS is not competitive compared with the cost-volume-based methods. One solution is to replace the $l \times l$ patch with the learned feature in Patchmatch. However, the accuracy of the learned photometric consistency in self-supervised stereopsis is still far from the traditional one. In future work, we plan to design a lightweight network architecture to represent not only the part correlation but also the pixel-wise wrapped image patches, jointly learned from a self-supervised learning-based strategy, which can meet the demand for more challenging tasks such as real-time MVS and dynamic MVS.

## 5   Conclusion

In this paper, we present a novel elastic part representation that encodes physically connected part segmentations with elastically-varying scales, shapes and boundaries. We build the correlation between the embedding space and the scene space to represent the surface connectedness and boundaries behind the image. A self-supervised MVS framework, ElasticMVS, is proposed to learn the representation and estimate per-view depth following a part-aware patchmatch-based algorithm. Benefiting from the inherent geometric correlations learned from the huge amount of images, ElasticMVS delivers state-of-the-art performance in both reconstruction completeness and accuracy, and extraordinary generalization ability in the challenging large-scale scenes.

## Acknowledgements

This work is supported in part by Natural Science Foundation of China (NSFC) under contract No. 62125106, 61860206003, 62088102 and 62171256, in part by Ministry of Science and Technology of China under contract No. 2021ZD0109901 and in part by Shenzhen Key Laboratory of next generation interactive media innovative technology, Funding No: ZDSYS20210623092001004.

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
