# Appendix

## A  Additional results

### A.1  Benchmarks

A detailed qualitative and quantitative comparison of state-of-the-art MVS methods on all scans in Tanks and Temples (*T&T*) dataset, as shown in Table 7, Table 8 and Fig.7. ElasticMVS achieves state-of-the-art performance among all supervised and self-supervised MVS methods and the tremendous performance improvement in *T&T advanced* benchmark, demonstrating the effectiveness and robustness of the proposed ElasticMVS to complex categories and geometries. The qualitative comparisons of all scenes in the *T&T* dataset also demonstrate that ElasticMVS precisely reconstructs the scenes with higher completeness and less noise.

Table 7: Quantitative evaluation of state-of-the-art learning-based methods on the Tanks and Temples (T&T) intermediate dataset [25].

|  | Method | Mean | Family | Francis | Horse | Lighthouse | M60 | Panther | Playground | Train |
|---|---|---|---|---|---|---|---|---|---|---|
| | MVSNet [47] | 43.48 | 55.99 | 28.55 | 25.07 | 50.79 | 53.96 | 50.86 | 47.90 | 34.69 |
| | CasMVSNet [15] | 56.84 | 76.37 | 58.45 | 46.26 | 55.81 | 56.11 | 54.06 | 58.18 | 49.51 |
| | UCSNet [10] | 54.83 | 76.09 | 53.16 | 43.03 | 54.00 | 55.60 | 51.49 | 57.38 | 47.89 |
| | PVAMVSNet [49] | 54.46 | 69.36 | 46.80 | 46.01 | 55.74 | 57.23 | 54.75 | 56.70 | 49.06 |
| Learning based | SurfaceNet+ [19] | 49.38 | 62.38 | 32.35 | 29.35 | 62.86 | 54.77 | 54.14 | 56.13 | 43.10 |
| Supervised | R-MVSNet [48] | 50.55 | 73.01 | 54.46 | 43.42 | 43.88 | 46.80 | 46.69 | 50.87 | 45.25 |
| | Point-MVSNet [7] | 48.27 | 61.79 | 41.15 | 34.20 | 50.79 | 51.97 | 52.38 | 43.06 |
| | PatchmatchNet [39] | 53.15 | 66.99 | 52.64 | 43.24 | 54.87 | 52.87 | 49.54 | 54.21 | 50.81 |
| | Patchmatch-RL [26] | 51.81 | 60.37 | 43.26 | 36.43 | 56.27 | 57.30 | 53.43 | 59.85 | 47.61 |
| | MVS$^2$ [11] | 37.21 | 47.74 | 21.55 | 19.50 | 44.54 | 44.86 | 46.32 | 43.48 | 29.72 |
| Learning based | M$^3$VSNet [16] | 37.67 | 47.74 | 24.38 | 18.74 | 44.42 | 43.45 | 44.95 | 47.39 | 30.31 |
| Self-supervised | SurRF [51] | 54.36 | **69.69** | 52.45 | 27.83 | **63.08** | **59.41** | **52.33** | 59.34 | 50.74 |
| | JDACS [43] | 45.48 | 66.62 | 38.25 | 36.11 | 46.12 | 46.66 | 45.25 | 47.69 | 37.16 |
| | **ElasticMVS (ours)** | **57.88** | 69.09 | **63.77** | **43.44** | 62.60 | 59.40 | 51.87 | **59.36** | **53.47** |

Table 8: Quantitative evaluation of state-of-the-art learning-based methods on the Tanks and Temples (T&T) advanced dataset [25], which demonstrates dramatic performance improvements w.r.t. state-of-the-art methods in both supervised and self-supervised approaches.

| Method | Mean | Auditorium | Ballroom | Courtroom | Museum | Palace | Temple |
|---|---|---|---|---|---|---|---|
| CasMVSNet [15] | 31.12 | 19.81 | 38.46 | 29.10 | 43.87 | 27.36 | 28.11 |
| R-MVSNet [48] | 29.55 | 19.49 | 31.45 | 29.99 | 42.31 | 22.94 | 31.10 |
| PatchmatchNet [39] | 32.31 | 23.69 | 37.73 | 30.04 | 41.80 | 28.31 | 32.29 |
| Patchmatch-RL [26] | 31.78 | **24.28** | 40.25 | 35.87 | 44.13 | 22.43 | 23.73 |
| **ElasticMVS (ours)** | **37.81** | 21.33 | **42.97** | **38.32** | **54.01** | **31.71** | **38.55** |

## B  Elastic part representation

### B.1  Training loss

Visualizations of the elastic part representation representation compared with variants trained without confidence map (w/o Con.) and spatial concentration loss (w/o Spa.), which are shown in Fig 8 and Fig 9. Our elastic part representation successfully recognizes the part on the same physical surface with elastically-varying scales, shapes and boundaries.

### B.2  Representation Comparison

We compare our elastic part representation with part representations in SLIC [2] and JDACS [43]. Since SLIC and JDACS both represent parts using labels that are different from our embedding-based representation, we highlight those pixels sharing the same label with the reference pixel denoted by the black "+". As shown in Fig 10, Our elastic part representations are more scalable in size while also better preserving object boundaries.

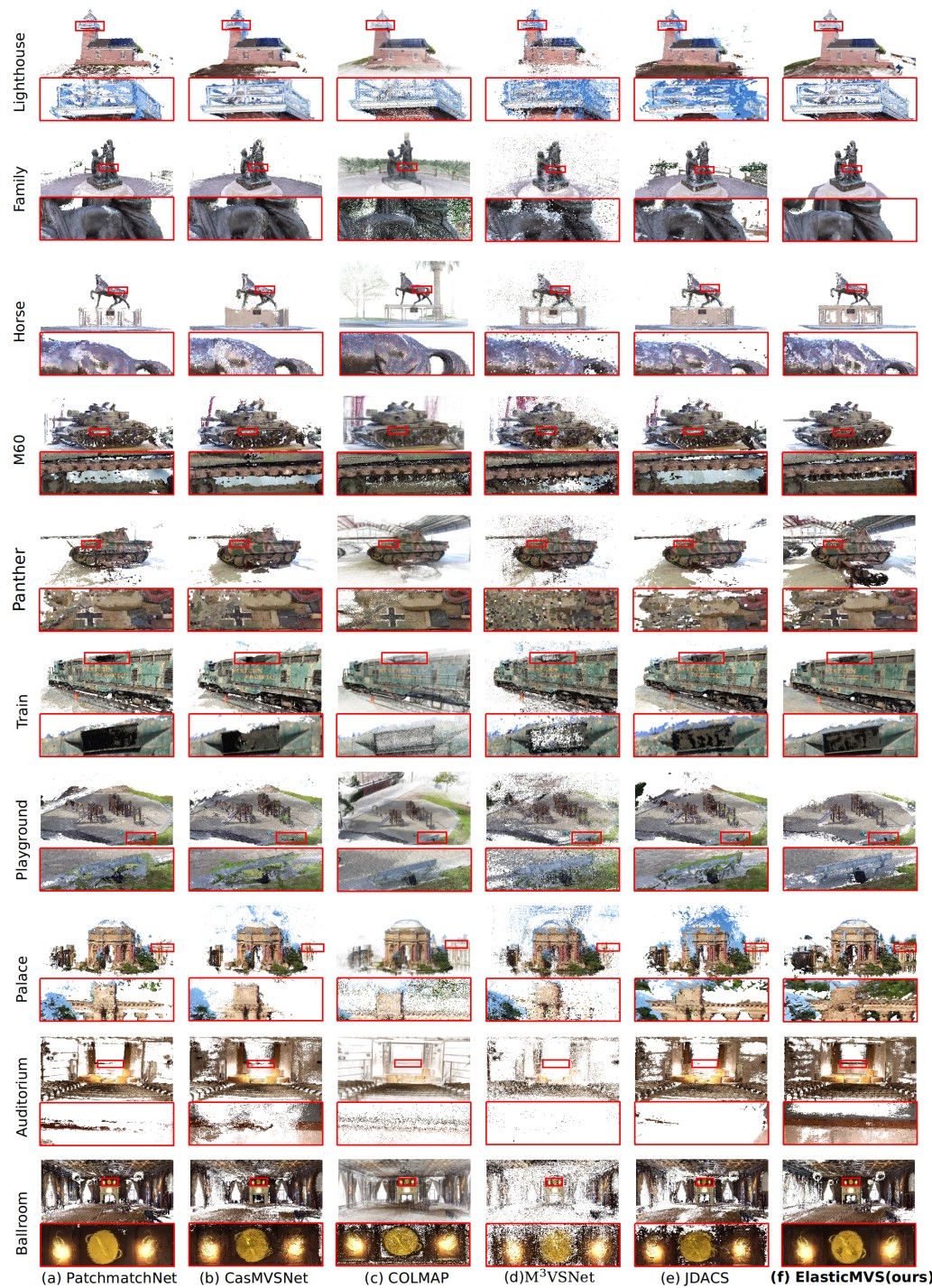

Figure 7: Point cloud visualization in the T&T intermediate and advanced benchmarks, compared with three types of reconstruction methods: supervised methods (a)(b) [39, 15], self-supervised methods (d)(e) [16, 43], and one traditional geometry-based method (c) [34].

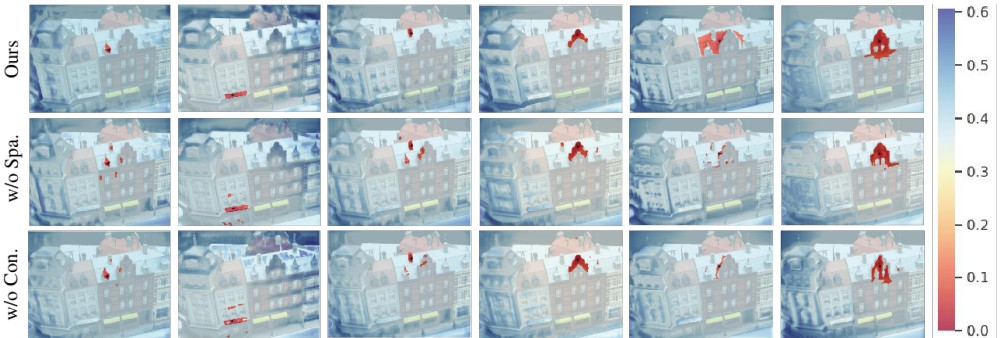

Figure 8: Visualization of the elastic part representation for DTU dataset. Part representations for specified pixels (black) are highlighted(red). Con. denotes confidence map, and Spa. denotes spatial concentration loss.

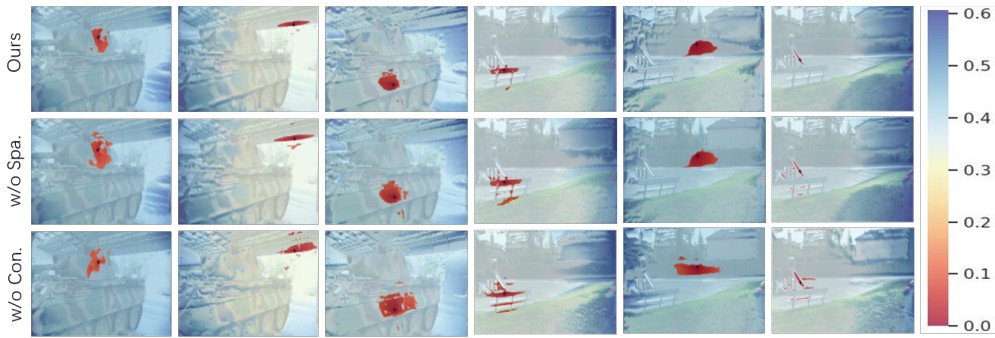

Figure 9: Visualization of the elastic part representation for T&T dataset. Part representations for specified pixels (black) are highlighted(red). Con. denotes confidence map, and Spa. denotes spatial concentration loss.

## C ElasticMVS

### C.1 Propagation

To better understand the effect of part-aware propagation, we visualize the sampled candidates for propagation (blue points) during 3 iterations of patchmatch given six pixels (red points), which are

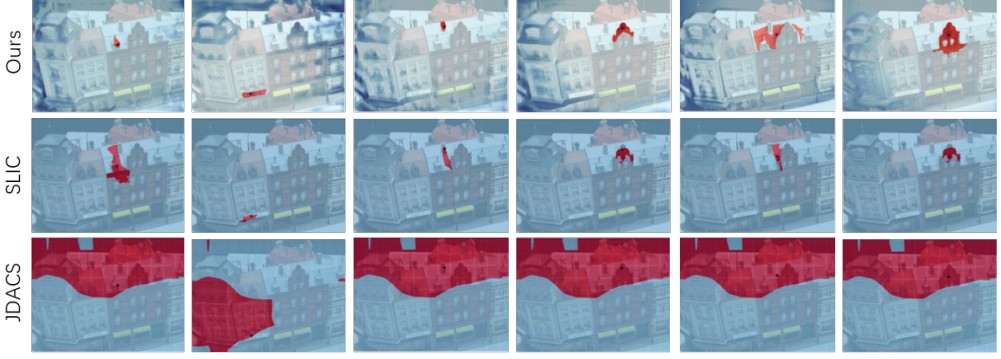

Figure 10: Visualization of part representations of different methods, pixels with the same label or the similar embedding as the reference pixel (denoted by black "+") are highlighted.

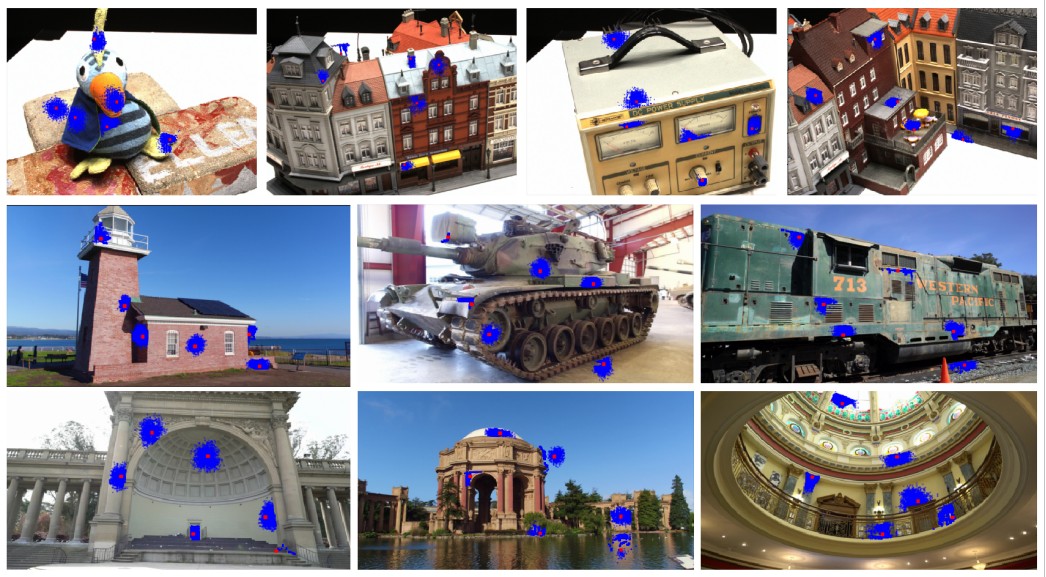

Figure 11: Visualization of the candidates (blue) during 3 patchmatch iterations and their corresponding target points (red).

shown in Fig 11. As expected, the receptive field for hypotheses propagation is elastically changed according to the the geometric shape and the texture pattern of the object.

## C.2    Number of iterations

The quantitative results with different number of hypothesis $(K_p, K_r)$ under each iteration described in Sec. 3.2 are summarized in Fig. 12. Obviously, the error converges after only 4 iterations of patchmatch using different number of hypotheses. Besides, using more hypotheses leads to faster convergence. Apart from that, our part-aware patchmatch is robust to the diverse initialization and local perturbation.

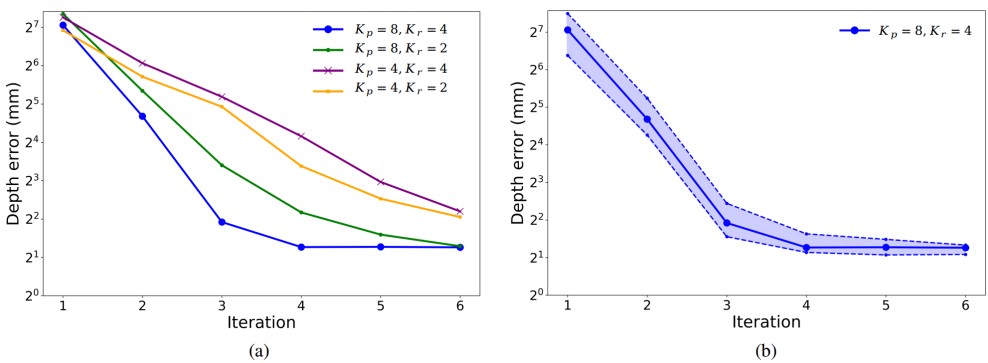

Figure 12: Performance under each iteration. (a) Different number of hypotheses $(K_p, K_r)$ for propagation and perturbation. (b) Error bars with the default setting of $K_p$ and $K_r$, which report the best and worst performances under different random seeds.

## C.3   Confidence map

The depth map predicted in each iteration is not always reliable, especially in textureless and high-reflectant areas. Therefore, we integrate multi-view geometric consistency constraints into the photometric consistency to increase the reliability and the accuracy of training and propagation. The visualization of each part is demonstrated in Fig. 13. We diminish reconstruction artifacts by firstly using a bilaterally weighted adaption of normalized cross correlation to measure the photometric consistency $c_p^{\text{pho}}$ between a reference view with the corresponding source views. After that, the geometric consistency $c_p^{\text{geo}}$ is computed as the forward-backward reprojection error between multiple views [34]. Besides, we compute surface smoothness $c_p^{\text{smo}}$ [35] to find smooth surface parts, which is the angle between surface normals of adjacent image points. This term identifies sudden changes of the surface normal direction at object edges [37], which is robust w.r.t. noisy normal vectors.

As shown in Fig. 13, for each reference image, the photometric consistency $c_p^{\text{pho}}$ (b), geometric consistency $c_p^{\text{geo}}$ (c) and surface smoothness $c_p^{\text{smo}}$ (d) for each pixel $p$ is separated computed. The final confidence $c_p$ (e) for training and propagation is the combination of these three factors: $c_p = c_p^{\text{pho}} \cdot c_p^{\text{geo}} \cdot c_p^{\text{smo}}$.

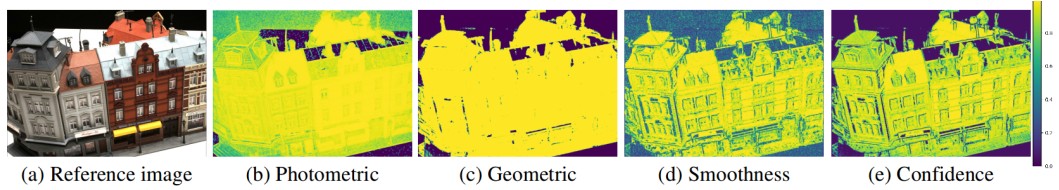

(a) Reference image     (b) Photometric     (c) Geometric     (d) Smoothness     (e) Confidence

Figure 13: Decomposition of the confidence map for training and candidate selection.