# OpenReview forum: "ElasticMVS: Learning elastic part representation for self-supervised multi-view stereopsis"
_NeurIPS.cc/2022/Conference — NeurIPS 2022 Accept_

### Official Review · Reviewer_hSNX · 2022-06-29

**Rating:** 7
**Confidence:** 5
**Soundness:** 4 excellent
**Presentation:** 3 good
**Contribution:** 4 excellent

**Summary:**

This paper proposes a self-supervised MVS framework that learns the representation and estimates depth following a part-aware propagation and evaluation scheme. To achieve this, this paper utilizes learns part-aware representation and optimizes a part-level consistency loss. The proposed method consists of part representation learning and part-aware matching. The part representation encodes geometric details to predict a piecewise-smooth depth map. The part-aware matching conducts the depth hypotheses propagation following the idea of patchmatch.

**Questions:**

1. How robust is the proposed method w.r.t. the threshold parameters?
2. Does the authors use the same threshold values for all datasets? or find the best parameters for each dataset?
3. Generally, how many seed points are obtained? (In figure 11 on supplementary materials, six pixels are provided.) Generally, six points are enough to cover all the images?

**Limitations:**

Yes

**Strengths And Weaknesses:**

Strength
1. The proposed method achieves state-of-the-art performance. The proposed method even outperforms some of the supervised methods.
2. In some datasets, the proposed methods show dramatic performance improvements w.r.t. the state-of-the-art methods in both supervised and self-supervised methods.
2. The proposed method does not require a fixed number of labels for segmentations and a ground-truth segmentation map.
3. The proposed method is geometrically reasonable and is quite novel.
4. The visualization of the elastic part representation demonstrates the effectiveness of the proposed method.

Weakness
1. The evaluation metric and unit are missing in table 3. Because DTU and T&T use different evaluation metrics, table 3 can be confused without the evaluation metrics.
2. The running time is not competitive compared to cost-volume-based methods.
3. The proposed method has many hyperparameters for the threshold, such as \epsilon, \xi, \eta, so I think it is quite vulnerable to the parameters. It would be better to show how robust the proposed method is w.r.t the hyperparameters.

---

> ### Author Response · Authors · 2022-08-01
> **Reply for details**
>
> We sincerely thank you for recognizing our contribution and for your valuable comments. In the following, we address your concern in detail.
>
> ## R4-1: Running time
> Since we inherit the conventional patchmatch-based technical route, the iterative propagation and evaluation process do have an inherent limitation in run time, which is due to the extremely time-consuming pixelwise differentiable warping process calculated multiple times. On the contrary, cost-volume-based methods only calculate homography once and then pass the warped features to the high-parral 3D CNN, they achieve a shorter run time. In the future, we plan to design a lightweight network architecture that uses an attention-based network that incorporates both the global structure correlation and the pixel-wise patchmatch process. Therefore, fewer computing iterations can be achieved while the superiority of part representation can be preserved.
>
> Nevertheless, our key contribution is a novel elastic part representation that builds the correlation between the latent space of the image representation and the scene space of the surface in a self-supervised way. We believe that our representation will help the community to bridge the gap between the multi-view stereopsis in computer vision and the point cloud reconstruction in computer graphics.
>
> ## R4-2: Robustness to the hyperparameters
> Thanks for your valuable suggestion and we have added a subsection in Sec. 4.2 of the final manuscript to discuss the robustness of the parameters. **Overall, our method is robust to the hyperparameters within a proper range.**
> Specifically, these hyperparameters can be categorized into scene-related (that is, depth range, image resolution) hyperparameters (i.e., $\epsilon$) and scene-irrelevant hyperparameters (i.e., $\xi$, $\tau$ and $\eta$). In our implementation, each scene-related hyperparameter should be modified according to the depth range of the dataset, while the non-geometry-related hyperparameters don’t need to be modified.
>
>
> ## R4-3: Experimental details
> ### R4-3-1: Missing unit
> Thanks for pointing out the omissions in our experiment. We will add the metric and unit in our final manuscript.
>
> ### R4-3-2: Meaning of the visualization in Figure 11
> Sorry for the misleading description, we’ll modify the caption of Fig. 11 in our final manuscript. In fact, the six red points are manually selected only for visualization. The real case is that every pixel in the image has a bunch of corresponding blue points, which represents the hypotheses selected during the past three iterations of patchmatch. We will modify the caption as follows: Visualization of the candidates (blue) during 3 patchmatch iterations and their corresponding target points (red).

---

### Official Review · Reviewer_cy7P · 2022-07-08

**Rating:** 5
**Confidence:** 3
**Soundness:** 3 good
**Presentation:** 2 fair
**Contribution:** 2 fair

**Summary:**

The paper proposed an interesting idea to improve the pixel-wise multi-view correspondence matching using geometric correlations.
An elastic part representation is proposed to encode the surface connectedness. Specifically, the input image is passed through a convolutional neural network to get the per-pixel elastic part representation. Then based on the estimated depth and normal map, the authors proposed self-supervised contrastive learning to encourage the part representation to be similar on the same physical surface.


**Questions:**

1) Please clarify why the idea of "propagation" is very similar to the "adaptive propagation" in PatchmatchNet while the latter is not suitable cited. Also, discussions on the difference between these two should be discussed.

2) Please clarify the usage of $z_p$ in the text.

3) What is the final form of the loss function used to train the CNN.

**Limitations:**

Well discussed.

**Strengths And Weaknesses:**

Strengths:

The authors presented a novel elastic part representation and provided an ablation study to prove the effectiveness.

A thorough experiment is conducted and the proposed method is compared to a large set of state-of-the-art methods, notable improvements are observed compared to other self-supervised methods.

Weakness:

Although contributions are presented in the paper, the core idea of the work is very close to PatchmatchNet (Wang et al. 2021a) and the resulting depth estimation relies on the initial depth estimation from Gipuma (Galliani et al. 2015). Especially the key idea of "Propagation" proposed in this paper is very similar to the "Adaptive Propagation" proposed in the PatchmatchNet (Wang et al. 2021a), more discussion on the differences in-between are expected.

Confusing symbols. $z_p$ in Sections 3.1 and 3.2 is referred to as "per-pixel elastic part representation" and "feature on pixel local p". I assume these two terms mean the thing?

The second paragraph of Section 3.1 is better fitted in the Network Training section and Equations (1) and (6) are the same (almost). Not sure why the same equation is written twice in the text.

The authors talked about the two loss functions used in the training but didn't provide what is the final loss function. I assume it is a weighted sum of these two loss functions?

In the Experiment Section, the values reported for Gipuma (Galliani et al. 2015) on the DTU dataset seem different from the ones reported in the original paper. Also, since the reconstruction results of COLMAP on T&T dataset are provided in Figure 4, it seems strange to not provide the quantitative evaluation in Table 2.

Nitpicking:

Abbreviations used without explanations, e.g. NCC

The supplementary document, Section C.3, first sentence:
"The depth map predicted in each iteration is not always unreliable ...", shouldn't it be "not always reliable"?

---

> ### Author Response · Authors · 2022-08-01
> **Reply for the main idea and other technical details**
>
> ## R3-1: Contribution and main idea
> Thanks for your in-deep review, but we disagree with the opinion that our idea of propagation is similar to “Adaptive propagation” in Patchmatchnet(Wang et al.) as follows:
>
> ### Difference between PatchmatchNet
> **Although PatchmatchNet can propagate hypotheses to neighbors adaptively based on Deformable Convolution Networks, the method lacks reliable and geometric-intuitive indicators on how to train and select the propagation candidates on the same surface.** Instead, candidates in our part-aware propagation are sampled based on the distance of each representation in the latent space, correlated to the scene space of the surface with a clearly defined loss function (Eq. 1 and Eq. 6).
>
> More specifically, propagation candidates in PatchmatchNet are directly predicted from a 2D CNN, a “black box”, on the reference feature map. **The network and the feature map are neither physically plausible nor geometrically intuitive to guarantee that the predicted candidates can lie on the same physically-connected surface.**
>
> More importantly, there are no explicit geometric constraints on the deformable propagation network during training. It is reasonable to doubt whether the network has learned reliable deformation. Furthermore, the whole training process in PatchmatchNet is supervised by the ground-truth depth map, which **cannot be implemented in the self-supervised task**.
> We have added the above description in the introduction and the related work of the final manuscript.
>
> ### Our major contribution
> The key idea of our work is actually a novel elastic part representation encoding physically-connected part segmentations, and we demonstrate that it can guide patchmatch propagation and evaluation to get better reconstruction.
>
> **Different from PatchmatchNet and all previous works in MVS, we aim to build the correlation between the latent space of the image representation and the scene space of the surface in a self-supervised way.** We designed a pixel-wise part representation that explicitly depicts surface correlations, i.e, the higher similarity between two representations directly denotes a higher probability they come from the same surface.
> **In fact, benefiting from the elastic part representation, the inherent geometric correlations can be effectively exploited by many other MVS methods.**
> For example, in CasMVSNet (Gu et al 2019), we can sample hypotheses more efficiently during the coarse-to-fine cascade volume generation by only sampling hypotheses in the same surface part. Besides, we can improve the final depth regression layer in MVSNet (Yao et al 2018) by our part-aware correspondence.
> **Overall, our major contribution lies in a new image representation and a new self-supervised framework, rather than a specific propagation implementation.**
>
> ### The effect of propagation
> As illustrated in Sec. 3.2, there are three major parts during the inference: 1. propagation, 2. part-aware correspondence, and 3.part smoothness. All three components contribute equally to the final result of the part-aware patchmatch, as demonstrated in Table 3. From the dramatic performance drop w/o PC, we can conclude that even with the adaptive propagation scheme, evaluating hypotheses using only photometric consistency can not achieve comparative performance. **Therefore, it is part representation, rather than propagation, that crucially improves the experimental results of the reconstruction.**
>
> **Statement on initialization:** Our depth prediction does not rely on the initial depth estimation from Gipuma. Instead, the depth map is initialized randomly within the range of interest. We only need an initialized depth prediction during the training stage of the elastic part representation.
>
>
> ## R3-2: Technical details
> ### R3-2-1: Final loss function
> The final loss function is the weighted sum of these two loss functions described as $L_{total}=L_c+\alpha_s L_s$, where $\alpha_s$ is a constant parameter which is set to 5*10^-3 in our implementation. We have added them to Sec. 3.4.
> ### R3-2-2: Difference between Eq. 1 and Eq. 6
> Since Eq.1 is the conceptual equation inherited from Soft-Nearest Neighbor Loss (Salakhutdinov and Hinton), which is the ideal equation used to describe the concept of our elastic part representation, it is not capable of the actual training process. Instead, Eq.6 describes a computable loss function with an efficient sampling strategy to approximate the property in Eq.1.
> ### R3-2-3: Description of $Z_p$
> We’ll modify them in the final manuscript. The two terms mean the same thing and we will replace “feature on pixel local p” with “per-pixel elastic part representation on p”.
>
>
> ## R3-3: Detailed questions
> Thanks for pointing out the omissions and mistakes, we have corrected them in the final manuscript.
> ### Values reported for Gipuma
> Our values reported for Gipuma actually comes from the previous MVS paper such as MVSNet (Yao et. al) and Patchmatchnet (Wang et al.).

---

### Official Review · Reviewer_pJ8f · 2022-07-09

**Rating:** 5
**Confidence:** 5
**Soundness:** 4 excellent
**Presentation:** 3 good
**Contribution:** 3 good

**Summary:**

This paper introduced elastic part representation into self-supervised MVS framework to achieve more accurate dense correspondences. The pixel-wise part representation is trained by a contrastive learning after a pre-training process. The proposed elastic part representation mainly identifies pixels of similar appearance for depth propagation. The proposed approach is evaluated on the DTU and T&T benchmark datasets and achieves promising results.

**Questions:**

1)	How was the initialisation of the depth map obtained for contrastive learning? Is this process repeated once an improved depth map is obtained?
2)	How were the depth hypotheses chosen? Are they related to the initialisation without the pixel-wise part representation learning?
3)	Please also clarify how the optimisation of the energy function in line 171 is solved. The reviewer interprets that no network learning is involved in the process listed in Figure.2.
4)	Does the runtime reported in Table 5 include the calculation of feature distance, confidence map and the normal map?


**Ethics Review Area:**

["I don’t know"]

**Limitations:**

1)	The method still cannot estimate accurate depth for textureless regions even with better propagation region estimation.
2)	It is potentially time consuming because it requires iterative optimisation, calculation of the feature distance, confidence map, and solve the optimisation problem. Surprisingly, the reported run time is reasonable.


**Strengths And Weaknesses:**

# Strengths
1)	This paper focuses on self-supervised learning for MVS. It adopts the contrast learning to identify pixels belonging to a connected surface or not based on learned features and guide the depth propagation
2)	It achieves promising results on two benchmark datasets.
3)	Comprehensive ablation studies are presented in the paper.

# Weaknesses.
1)	The figure 1 cannot accurately reflect the proposed algorithm. It is a multiple stage framework. Especially, the proposed framework relies Gipuma without part representation to obtain an initial depth which will be used to define the positive and negative set for contrastive learning for the start of the algorithm. It is missing from Fig.1.
2)	Some arguments in the paper are kind of misleading. Line 29-31 argues about the pixel-wise photometric regularisation is sensitive to textureless regions or illumination changes. It is not clear how the proposed elastic parts representation could overcome this. My understanding is the contrastive learning is based on the positive and negative samples obtained from an unsupervised learning framework which is already affected by the textureless and illumination changes. No depth information can be obtained from this initialisation. It is questionable the proposed representation can fix wrong depth estimation for the textureless regions. Line 32-35, It is not clear why these are related to the reconstruction.
3)	Depth estimation is obtained by traditional optimisation rather than learning based on the description between line 166-171.

---

> ### Author Response · Authors · 2022-08-01
> **Reply for the depth recovery and other technical details**
>
> We sincerely thank you for recognizing our contribution. In the following, we address your concern in detail.
> ## R2-1: The connection between the part representation and the depth prediction
> Sorry for the confusion and we have clarified their connection in the introduction and related work of the revised manuscript.
>
> We aim to build the correlation between the latent space of the image representation and the scene space of the geometry in a self-supervised way, which also provides a neat solution to bridge the gap between the multi-view stereopsis and the point cloud reconstruction. Therefore, the purpose of our part representation is not to directly fix the wrong depth, but to provide the inherent geometric correlations such as surface connectedness, smoothness and boundaries to guide the part-aware patchmatch and depth prediction. We have demonstrated that it can guide patchmatch propagation and evaluation to get better reconstruction.
> **In R1-2, we have detailedly explained the strategy to learn a noise-robust part representation with clear boundaries and complete part segments given only incomplete and noisy depth maps.**
> ### R2-1-1: How to fix the wrong depth?
> **Furthermore, given a part representation or propagation region estimation, there are several ways to estimate more accurate and complete depth predictions.** Previous work in MVS solved the problem by introducing pre-defined global structural priors such as planes to resolve the ambiguity (Romanoni and Matteucci, 2019; Bódis-Szomorú et al., 2014). For instance, TAPAMVS(Romanoni et al.) segments the whole image into a bunch of superpixels and assumes textureless areas piecewise planar. RANSAC is applied to estimate its corresponding plane parameters. Beyond that, a bunch of works in point cloud reconstruction [2] deal with reconstruction artifacts with outlier, noisy or missing data. For instance, implicit representation [3] is used to provide global surface smoothness prior given noisy data, and exterior visibility [4] is used to detect and eliminate redundant data. However, neither the handcrafted plane structure nor the complex smoothness prior can provide sufficiently diverse and accurate surface part modeling for real-world scenarios.
>
> **In ElasticMVS, we provide a novel solution, L1 median loss, to encourage a non-local piecewise smoothness surface prediction in each part segment.** Several works (Lipman et al.) have demonstrated that optimizing the L1 median loss is robust to the outliers and gives a piecewise second-order surface approximation if the underlying surfaces are physically connected.
>
> [2]Berger, Matthew et al. “State of the Art in Surface Reconstruction from Point Clouds.”  Eurographics(2014).
>
> [3]Kazhdan, Michael M. et al. “Poisson surface reconstruction.” SGP(2006).
>
> [4]Katz S, et al. Direct visibility of point sets. SIGGRAPH 2007.
>
> ## R2-2: Technical details
> ### R2-2-1: Initialization
> As described in Sec. 3.3, from the very beginning, the depth and normal are randomly initialized under the uniform distribution. After that, a depth map (using Gipuma) and a confidence map (using COLMAP) are passed to the contrastive training process.
> In this paper, we did not repeat the training process using the improved depth map since we found that the part representation has achieved favorable results using only the initial depth estimation.
> In future work, we consider building a lightweight network architecture that incorporates global part and local patch representations to replace the laborious and time-consuming patchmatch process. At that time, a refined part representation can be updated immediately once an improved depth map is obtained, without the need for depth map initializations.
> ### R2-2-2: Depth hypotheses
> Different hypotheses choosing strategies are utilized before and after the training process. Before the training process, we inherit the strategy of Gipuma(Galliani et al.). After the training process, we choose hypotheses based on the feature correlation of nearby pixels as illustrated in Fig.3.
> ### R2-2-3: Optimization
> Indeed, the function is not directly solved using optimization mechanisms such as gradient descent, but rather approximated using a discrete sampling strategy.  This strategy has long been proposed in the field of stereo-matching (Schönberger et al., Galliani et al., Xu et al.).
> ### R2-2-4: Runtime
> Every calculation process has been counted. The calculation of feature distance is only an operation of the inner product, which is faster than calculating patch correspondence (which involves differentiable homography). Moreover, the optimization is solved by the efficient patchmatch, which is relatively a lightweight solution compared to cost-volume-based methods (Wang et al.). Besides, the confidence map is a simple "loss function" and the normal map can be obtained from a simple convolution layer.
> ## R2-3: Teaser figure
> We have added the illustration of the initialization step in the final manuscript.

---

### Official Review · Reviewer_HBPd · 2022-07-12

**Rating:** 5
**Confidence:** 4
**Soundness:** 2 fair
**Presentation:** 3 good
**Contribution:** 2 fair

**Summary:**

This paper proposed a new self-supervised MVS method called ElasticMVS. Instead of relying on local photometric consistency as in existing methods, the proposed method utilizes geometric correlations parsed from images as guidance for pixel-wise multi-view correspondences. The paper first proposed a novel elastic part representation that encodes spatial proximity and surface connections, which is trained by a contrastive learning-based strategy. The paper then proposed a self-supervised MVS framework namely ElasticMVS that estimates per-view depth using a part-aware propagation and evaluation scheme. Experiments show that ElasticMVS archives significant performance gain over both the supervised and self-supervised approaches.

**Questions:**

- What exactly is $\beta$ in equation (7)? It is not mentioned in the implementation details.

Minor:
Line 94: "sucn" -> "such"

**Ethics Review Area:**

["I don’t know"]

**Limitations:**

Yes.

**Strengths And Weaknesses:**

Strengths:
-  Addressing the limitation of local photometric consistency is an important topic for self-supervised MVS. The insights and motivation of the proposed method are good.
- The paper presents extensive and sufficient comparisons with baselines, and provides detailed and informative ablation studies.
- The proposed method works very well and achieves state-of-the-art performance, supported by various qualititive and quantitative results.

Weaknesses:

I don't have major concerns about this paper but I have a few questions to ask for clarification.
- It is not clear to me why it is necessary to learn the part representation $z_p$. $z_p$ is learned primarily using geometric cues such as the 3D point-to-plane geometric distance, which is also available during inference. Why can't we design some simpler similarity functions directly based on the supervision signal of $z_p$ which is the 3d distances between p and q? For propagation, the candidates for pixel p can just be its nearby pixels that satisfy line 113-117. What is the benefit of learning the part representation compared to directly using geometric based heuristics?
- The quality of the elastic part representation relies on initial depth map. How robust is the elastic part representation to the noise and error in the initial depth map? Is the method still able to handle textureless regions if the initial depth map for these regions are already erroneous?
- To me, the abstract is a bit overclaiming. In line 6 it says the method explores "geometric correlations such as surface connectedness and occlusion boundaries". This sets my expectation high but I found it does not really match the proposed method. The method does not reason about surface connectedness or occlusion boundaries explicitly, but rather implicitly. In my opinion the key part of the method  is clustering points based on geometric proximity (both 3D point-to-plane and in full 3D), and use it to guide patchmatch propagation and evaluation. It is still a neat idea but the description in the abstract can be a little misleading. I think the sentence in line 6 can be replaced with a more proper description.
- It is possible that two spatially far apart pixels can be coplanar and thus have a small 3D point-to-plane distance, but that is not desirable. What part of the method suppresses this kind of points?

---

> ### Author Response · Authors · 2022-08-01
> **Reply for the necessity of the representation, training robustness and other details**
>
> Dear reviewer, we sincerely thank you for your in-deep comments and for recognizing the novelty of our method. In the following, we address your concern in detail.
>
> ## R1-1: Necessity of the representation
> Thanks for your valuable comment and we have clarified it in the revised manuscript. In our opinion, there are three major reasons to learn the part representation z_p:
>
> **Firstly, the geometric cues are usually unavailable during predictions.** For instance, there are no confident depth predictions in textureless regions, where physically-connected boundaries have to be guessed during inference. Therefore, a heuristic geometric regularization suffers from over-smoothed surfaces and rough boundary predictions. As Table 3 demonstrated, a naive surface de-noising and completion implementation still suffer from low accuracy and noise.
>
> Besides, the part-level representation for reconstruction requires high-precision geometry, which is obtained from either coarse-to-fine refinement or several stages of propagation from random initialization. **These processes are time-consuming compared to a direct forward pass of a 2D ConvNet in ElasticMVS.**
>
> Furthermore, Since we are aiming at the self-supervised task, the only way to gain the geometry information is to utilize the cues from photometry. **However, the geometry recovered from pure photometric consistency always suffers from incomplete and inaccurate surface predictions.** We have proposed several ways to robustly train the elastic part representation z_p with the noise and error in the initial depth map, as described in Sec. 3.3 and R1-2.
>
> Overall, the necessity to learn the part representation instead of directly using geometry heuristics is as follows:
>
> • **Accessibility**: The geometry information gained from any photometric loss is usually missing in some areas of the image, especially in textureless regions. A heuristic geometry regularization is unavailable in these regions.
>
> • **Speed**: During inference, the scales, shapes and boundary information is obtained from a 2D ConvNet without laborious optimizations.
>
> • **Robustness**: The geometric-based heuristics from photometric loss such as the point-to-plane distance are unreliable, especially in textureless and illuminant regions. However, it can be solved during the training stage with proper training losses and strategies.
>
>
> ## R1-2: Training robustness
> **During the training stage, we have properly dealt with the noise and error in the initial depth map using the confidence map.** Followed by COLMAP, the confidence map (line 137 and Fig. 13), a combination of photometric consistency and geometric regularization, is used to identify and diminish reconstruction artifacts. Regions with low confidence contain noisy photometric predictions, textureless regions and sudden changes in the surface normal.
>
> **More importantly, the contrastive (Eq. 6) and concentration loss (Eq. 7) are both designed to deal with training in low-confident regions.** Specifically, contrastive loss (Eq. 6) only acts on pixels with above-the-threshold confidence, which prevents the reconstruction artifacts from ruining the contrastive learning process, while concentration loss described in Eq.7 acts on all pixels, and encourages the spatially concentrated representation.
>
> To take it a step further, the training process can be seen as a "propagation scheme" that propagates the correct representation distance from high confidence regions to low ones. Thus, the embedding distance trained in high-confident regions could be successfully passed to low-confident ones.
>
> As shown in Fig. 6 and 8, embeddings from the same surface successfully gather together in the latent space in textureless regions.
>
> We will clarify them in Sec.3.3 of the final manuscript.
>
> ## R1-3: Abstract
> Thanks for your valuable suggestion. To make it clearer, we will modify part of the abstract (line 6) in the final manuscript as follows:
> In this paper, we show that geometric proximity such as surface connectedness and occlusion boundaries implicitly inferred from images could serve as reliable guidance for pixel-wise multi-view correspondences.
>
> ## R1-4: Training spatially far apart pixels
> Fig.6(b) shows that simply applying contrastive loss (Eq. 6) does lead to the spatially separated coplanar points being close in the latent space. Here's one reason why the concentration loss (Eq.7) comes in. The spatial concentration loss punishes the case when spatially far apart pixels have similar embeddings [1], therefore encouraging the embeddings of spatially distant points to be isolated.
>
> [1] Yang, Fengting et al. “Superpixel Segmentation With Fully Convolutional Networks.” _(CVPR) (2020)_.
>
>
> ## R1-5: Minor questions
> Thanks for pointing out the mistakes and we will correct them in the final manuscript. Specifically, $\beta$ is a constant parameter that controls the weight of the feature similarity between p and q, which is set to 0.5 in our implementation.

---

### Author Response · Authors · 2022-08-01
**Major contribution and technical details**

We thank all the reviewers for their thorough reviews and for appreciating the novelty of our method. We have highlighted the corresponding changes in the revised manuscript. We look forward to the discussion with all the reviewers. In the following, we will give our responses to the major issues mentioned by the reviewers.

# 1. Reply for the major contribution
## 1.1 Major contribution
Our key contribution is a novel elastic part representation that builds the correlation between the latent space of the image representation and the scene space of the surface in a self-supervised way, which is different from the previous MVS or semantic learning methods that only focused on improving the prediction quality separately through supervised training. Besides, we demonstrate that building the correlation can guide patchmatch propagation and evaluation to get better reconstruction. **We believe that our representation will help the community to bridge the gap between the multi-view stereopsis in computer vision and the point cloud reconstruction in computer graphics.**

## 1.2 The necessity of our part representation
The purpose of our part representation is to provide the inherent geometric correlations such as surface connectedness, smoothness and boundaries to guide the part-aware patchmatch and depth prediction. A heuristic geometric regularization is usually unavailable during predictions. For instance, there are no confident depth predictions in textureless regions. Therefore, directly applying geometric regularization often suffers from over-smoothed surfaces and rough boundary predictions. Besides, the part-level representation for reconstruction requires high-precision geometry, which can not be easily obtained in a self-supervised manner.
Since it is important to learn a noise-robust part representation with clear boundaries and complete part segments, we have proposed several ways to robustly train the elastic part representation with the noise and error in the initialization.

# 2. Reply for technical details
## 2.1 Training robustness w.r.t initialization
**During the training stage, we have properly dealt with the noise and error from the initialization.** Followed by COLMAP, the confidence map, a combination of photometric consistency and geometric regularization, is used to identify and diminish reconstruction artifacts. Regions with low confidence that contain noisy photometric predictions, textureless regions, and sudden changes in the surface normal are ignored from the contrastive learning process.
On the other hand, the contrastive (Eq. 6) and concentration loss (Eq. 7) are both designed to deal with training in low-confident regions. Therefore, there are three goals in the two loss functions: 1. In high-confident regions, we compact the representation in parts with close surface distance; 2. In high-confident regions, we separate the representation lies on different surface parts; 3. In low-confident regions, we encourage the representation to be as close as possible to nearby features from high-confident regions. Thus, the embedding distance trained in high-confident regions could be successfully passed to low-confident ones.
## 2.2 Speed
Since the whole optimization process during inference is inherited from the conventional patchmatch-based technical route, it has the potential to optimize efficiently by sampling a discrete set of hypotheses and choosing the best among them (see  Patchmatchnet(Wang et al.)).
Besides, the calculation of the feature distance, the confidence map and the normal map can be achieved by simple and efficient operations. However, the speed of ElasticMVS is majorly constrained by the iteratively time-consuming pixel-wise differentiable warping process. In the future, we plan to design a lightweight network architecture that incorporates the part correlation into the pixel-wise patch match process. Therefore, fewer computing iterations can be achieved while the superiority of part representation can be preserved.

---

### Meta-Review · Area_Chair_4B4q · 2022-08-26

**Recommendation:** Accept
**Confidence:** Less certain

**Metareview:**

All the reviewers acknowledged the strength of the paper: self-supervised learning for MVS using contrastive learning to help correspondence based on learned features, SOTA results are obtained on DTU and T&T benchmarks, and the evaluations/ablation studies are well presented. The reviewers also shared weaknesses: the elastic part representation may not deal with textureless regions, the core idea of “propagation” in the paper is very close to PatchmatchNet and too many parameters for threshold. The reviewers engaged in the rebuttal and discussion phase, and they all decided to keep their ratings. In general, the proposed part representation is interesting and leads to promising results. Please address the issues of how the learned representation can fix the missing depth regions caused by texturelessness as pointed out by both reviewer pJ8f and HBPd.

**Award:**

No

---

### Decision · Program_Chairs · 2022-09-14

Accept